# Determination of Cadmium (II) in Aqueous Solutions by In Situ MID-FTIR-PLS Analysis Using a Polymer Inclusion Membrane-Based Sensor: First Considerations

**DOI:** 10.3390/molecules25153436

**Published:** 2020-07-29

**Authors:** René González-Albarrán, Josefina de Gyves, Eduardo Rodríguez de San Miguel

**Affiliations:** Departamento de Química Analítica, Facultad de Química, UNAM, Ciudad Universitaria, 04510 Cd. Mx., Mexico; renegalbarran@comunidad.unam.mx (R.G.-A.); degyves@unam.mx (J.d.G.)

**Keywords:** cadmium (II), polymer inclusion membrane, FTIR, chemometrics, PLS

## Abstract

Environmental monitoring is one of the most dynamically developing branches of chemical analysis. In this area, the use of multidimensional techniques and methods is encouraged to allow reliable determinations of metal ions with portable equipment for in-field applications. In this regard, this study presents, for the first time, the capabilities of a polymer inclusion membrane (PIM) sensor to perform cadmium (II) determination in aqueous solutions by in situ visible (VIS) and Mid- Fourier transform infrared spectroscopy (MID-FTIR) analyses of the polymeric films, using a partial least squares (PLS) chemometric approach. The influence of pH and metal content on cadmium (II) extraction, the characterization of its extraction in terms of the adsorption isotherm, enrichment factor and extraction equilibrium were studied. The PLS chemometric algorithm was applied to the spectral data to establish the relationship between cadmium (II) content in the membrane and the absorption spectra. Furthermore, the developed MID-FTIR method was validated through the determination of the figures of merit (accuracy, linearity, sensitivity, analytical sensitivity, minimum discernible concentration difference, mean selectivity, and limits of detection and quantitation). Results showed reliable calibration curves denoting systems’ potentiality. Comparable results were obtained in the analysis of real samples (tap, bottle, and pier water) between the new MID-FTIR-PLS PIM based-sensor and F-AAS.

## 1. Introduction

Heavy metals are persistent toxic metals or metalloids present at low concentrations, in all parts of the environment. Human activity affects the natural geological and biological redistribution of these metals through pollution of air, water, and soil. Therefore, highly accurate and sensitive spectrophotometric and spectrometric analytical techniques are used for their measurement in complex matrices, e.g., atomic absorption (Flame atomic absorption spectroscopy (F-AAS), Graphite furnace atomic absorption spectroscopy (GF-AAS), Hydride generation atomic absorption spectroscopy (HG-AAS), and Cold vapor atomic absorption spectroscopy (CV-AAS)), emission (Inductively coupled plasma optical emission spectroscopy (ICP–OES)), and mass (Inductively coupled plasma mass spectrometry (ICP-MS)) methods. At present, analytics and environmental monitoring are among the most dynamically developing branches of chemical analysis. The general trends in both areas can be classified into two basic groups—(i) development of new methodical procedures, and (ii) new achievements in the construction of measuring instruments (instrumentation) [1]. Examples of the first group developed in the pursuit of obtaining complex information on environmental quality are the introduction of solventless techniques to the analytical practice for sample preparation and multidimensional techniques, while examples of the second are the design of new sensors and detectors for conducting measurements in situ. The general trend in analytical instrumentation toward a smaller size, improved reliability, and easy operation, now makes it possible to count with portable UV-VIS, FTIR, Raman, and NIR spectrometers [2,3]. Successful applications of such technology in process automation [4], chemical reaction monitoring [5], homeland defense [6], food [7,8], controlled drugs [9], artwork [3,10] and hazardous chemical [11] analyses were reported. However, difficulties related to the production of large amount of information and a lack of selectivity and compromised detection capabilities must be handled. As for the first ones, the use of multivariate mathematical and statistical (chemometric) techniques can be profitably exploited. As for the second ones, adequate sample preparation methods might be employed. Due to the growing demand for techniques involving fewer toxic reagents, less time-consuming protocols, with lower limits of detection, facility of sampling and elimination of interferences, simple preparation methods that can be directly coupled with the measurement technique have a great potential in environmental analysis. Due to its easy implementation, simplicity of operation, selectivity, stability, versatility, and minimal power consumption, liquid membrane-based sample preparation and preconcentration techniques have gained growing attention [12,13]. Polymer inclusion membranes (PIM) are a kind of liquid membrane that is adequate for separation and preconcentration [14] of analytes, with the additional advantages of an easy synthesis and the possibility to perform in-situ metal analysis by X-ray [15], VIS [16,17,18], and fluorescence [19] spectroscopies, including its use in a continuous flow system [20]. Promising applications of PIMs as immunosensors for *Salmonella typhimurium* [21], for the detection of chlorpyrifos, diazinon, and cyprodinil in natural waters samples [22], and for flow injection determination of V(V) [23], and thiocyanate [24] were recently reported. Lately, the increasing interest in these membranes in analytical chemistry was reviewed, as they were adapted to new and novel applications [25]. 

In this work, a simple multivariate sensor for measuring cadmium (II) in waters employing a PIM was developed and characterized. Direct analyses on the membrane were conducted using VIS and MID-IR spectroscopic techniques. The partial least squares (PLS) chemometric algorithm was applied to quantitatively measure the amount of metal in the membrane. To the best of our knowledge this is the first study in which simultaneous analyses in the VIS and MID-IR spectral regions were performed in a PIM for quantitative reasons, and multivariate regression was applied to such a purpose. The work showed that the results obtained by the validated MID-FTIR-PLS PIM-based sensor compared well to those generated by (F-AAS), thus the new sensor could be quite suitable for on-site analyses with portable equipment.

The FTIR is an analytical technique based on the interaction of IR radiation and a molecule. It is well-known that IR is divided into three regions—near (NIR-IR), mid (MID-IR), and far (F-IR). MID-IR and NIR-IR are non-destructive, fast, repeatable and cost-effective techniques. In the MID-IR region, absorptions are generated by overtones and fundamental vibrations of the –CH, –NH, –OH groups, among others [26]. The MID-IR region can lead to quantitative analysis since the absorbance of the sample is proportional to the number of functional groups. In contrast, NIR-IR is characterized by the presence of broad bands and overtones, making this region less useful for univariate quantitative analysis; nevertheless, the significant differences in positions of functional groups also provide a source of information [27]. A useful way to solve the spectral overlapping and to use all the information contained in the spectrum of complex matrices is the multivariate calibration methods [28], extensively applied to NIR-IR [29,30,31,32,33]. Several works demonstrated that NIR-IR techniques can be applied as a tool for quantitative multivariate analysis, especially with the combination of separation and preconcentration methods for organic compounds [34,35], and metal ions [36,37,38], where some of them use NIR-IR followed by MID-IR and Raman [39]. Even though it’s a drawback, the popularity of NIR is related to the good band assignments, improvements in instrumentation, and progress in statistical and mathematical methods [39]. In some reports, the capabilities of NIR-FTIR and MID-FTIR in the analysis of metals in different matrices were compared, and different advantages and constrains were found between them [27,40,41].

## 2. Results and Discussion

### 2.1. Establishment of Liquid-Solid Extraction Conditions

PIM composition was selected according to Aguilar et al. [42], i.e., (23.4 ± 0.2)% CTA, (54.5 ± 0.2)% NPOE, and (21.1 ± 0.4)% (*w*/*w*) Kelex 100. With this composition (97.2 ± 0.4)% of the metal was extracted in 4 h from an initial 1 × 10^−4^ mol dm^−3^ solution. Extraction experiments at 20, 40, 60, 120, 180, and 240 min showed that after 40 min, equilibrium in the system was reached. Further experiments were then performed using 60 min of extraction time and 1 × 10^−4^ mol dm^−3^ cadmium (II) solutions. The pH study was evaluated in the range 4 to 9. In Figure 1, the characteristic increase in extraction percentage with pH of acid extractants like 8-hydroxyquinoline, was observed [43]. In the higher pH region, the extraction percentage becomes inversely proportional to the hydrogen ion concentration, due to an increase in the concentration of the dissociated acid form of the extractant, L^−^, while in the lower pH region, an opposite behavior was observed due to the predominance of the acid form, HL, which did not favor the extraction. At pH ≥ 8, an extraction percent of at least (97.7 ± 0.2)% was achieved. Using the MEDUSA software [44], the predominance of the free ion species in the medium was evidenced below pH = 8. Above this value, the presence of hydroxide complexes and the Cd(OH)_2_ precipitate became relevant, and for this reason higher pH values were not further employed.

As for the influence of the initial cadmium (II) content, C_o_, metal concentrations were varied in the following sequence: 5 × 10^−3^, 1 × 10^−3^, 7.5 × 10^−4^, 5 × 10^−4^, 4 × 10^−4^, 2.5 × 10^−4^, 1 × 10^−4^ mol dm^−3^. Figure 2 shows the remaining aqueous equilibrium metal concentration. It was observed that for high concentrations (7.5 × 10^−4^, 1 × 10^−3^, 5 × 10^−3^ mol dm^−3^), less than 10% of the metal was extracted; in contrast, at about 5 × 10^−4^ mol dm^−3^, 50% was extracted, and at 1 × 10^−4^ mol dm^−3^, (97.6 ± 0.2)% of the metal was retained in the membrane phase.

It is interesting to note that from 1 × 10^−3^ mol dm^−3^ and on, the metal was practically not extracted in the system, due to a saturation phenomenon of the extracting phase. Further experiments were then performed, maintaining cadmium (II) concentrations within the range of 5 × 10^−4^ to 1 × 10^−4^ mol dm^−3^. In addition, the characterization of the systems in terms of its absorption capacity was studied.

### 2.2. Adsorption Isotherm

The amount of metal in the membrane phase, *q_e_* [mmol g^−1^] was plotted with respect to its equilibrium concentration in the aqueous phase, C_e_ [mmol cm^−3^] (Figure 3A), for a range of aqueous concentrations from 6.94 × 10^−7^ to 3.82 × 10^−4^ mol dm^−3^. As a Langmuir type isotherm was observed Equation (1), linearization of the data *(C_e_/q_e_* = f(*C_e_*)) was applied to determine the adsorption constant, *K_L_* (cm^3^ mmol^−1^), and the maximum adsorption capacity, *q_max_* (mmol g^−1^), parameters.
(1)$$q_{e}=\frac{q_{max}K_{L}C_{e}}{1+K_{L}C_{e}}$$

After performing the analysis, a careful inspection of the data pointed toward two different regions, depending on the metal content, high (1.63 × 10^−6^ to 5.16 × 10^−5^ (mmol cm^−3^)) and low (1.11 × 10^−6^ to 9.33 × 10^−7^ (mmol cm^−3^)), in which the degree of fit of the model was better performed, assuming different model parameters for each region (Figure 3B). 

In Table 1, both sets of model parameters are reported. It can be observed that *q_max_* decreased with a diminishing *C_e_*, as it probably became independent of the metal content [45], increasing the data dispersion. This behavior was further analyzed through computation of the separation coefficient, *R_L_*, defined by
(2)$$R_{L}=\frac{1}{1+K_{L}C_{O}}$$

Values of *R_L_* > 1 were indicative of a non-favorable adsorption; *R_L_* = 1 indicates a linear adsorption, while 0 < *R_L_* < 1 were observed in favorable adsorption. An irreversible adsorption was present in such a case where *R_L_* = 0 [46], in the high concentration range of the studied PIM system 0 < *R_L_* < 1, pointing out a favorable adsorption [47]. On the contrary, in the low range, *R_L_* ≈ 0, denoted an irreversible adsorption [46]. This observation agreed well with the two distinct regions observed in the adsorption isotherm. Comparing *q_max_* to other cadmium (II) sorbents, Fan et al. reported *q_max_* = 0.545 mmol g^−1^ for *P. Simplicissimum* [48], while Chakravarty et al. reported *q_max_* = 0.0948 mmol g^−1^ for *Areca catechu* [49], and Singh et al. reported *q_max_* = 9.43 × 10^−4^ mmol g^−1^ for *Trichoderma viridae* [50]. Some inorganic sorbents like activated alumina CNT nanoclusters, oxidized CNTs, and Fe_3_O_4_@TA showed maximum cadmium (II) adsorption capacities of 229.9, 11.01, and 286 mg g^−1^ [51] (equivalent to 2.04, 0.098, and 2.54 mmol g^−1^, respectively). These results showed that a wide interval of maximum sorption capacities for the metal could be obtained, depending on the type of sorbent, pH, temperature, ionic force, among other factors. The obtained *q_max_* values then lies within the reported ranges.

### 2.3. Enrichment Factor

The enrichment factor, *E*, defined by
(3)$$E=\frac{{\left[Cd\left(II\right)\right]}_{membrane}}{{\left[Cd\left(II\right)\right]}_{initial}}$$
is a measure of the preconcentration efficiency of the system. When plotting ${\left[Cd\left(II\right)\right]}_{membrane}$ (mmol g^−^^1^) as a function of ${\left[Cd\left(II\right)\right]}_{initial}$ (mmol dm^−^^3^) within the interval 6.94 × 10^−7^ a 3.82 × 10^−4^ mol dm^−3^ a linear relationship was observed, denoting a constant value of this parameter. From the slope, a value of *E* = 29.2 was determined. This result guaranteed the application of the sorption in the PIM as an adequate preconcentration method. The initial amount of cadmium (II) in the aqueous phase could be predicted from the amount of cadmium (II) in the membrane phase and the constant value of *E* in the metal concentration range, in which this constant value was attained. From the comparison of the *E* value with other membrane-based cadmium (II) preconcentration methods, the PIM once again presented an acceptable intermediate value, as Castro et al. reported *E* = 17.9, with the use of liquid membranes containing 2-APHB in toluene as extractant [52], while Peng et al. reported *E* = 387 using hollow fibers with dithizone dissolved in a mixture of 1-octanol and oleic acid [53]. Evidently, the *E* value should be dependent on the type of sorbent, cadmium (II) content, pH, temperature, ionic force, among other factors.

### 2.4. Stoichiometry of the Extracted Complex

Experiments in which Kelex 100 concentration was varied (1.9, 2.8, 3.6, 4.6, 7.1, 9.3, 10.9, and 12.3 *w*/*w*%), maintaining constant amounts of CTA and NPOE, were performed to determine the stoichiometry of the extracted complex through conventional graphical slope analysis. According to Aguilar et al. [42], cadmium (II) extraction with Kelex 100 proceeded through the reaction
(4)$$Cd^{2+}+n\bar{HL}+NO_{3}^{-}\overset{\;}{\to }\;\bar{CdH_{n-1}L_{n}NO_{3}}+H^{+}$$ in which $\bar{HL}$ stands for the extractant, $\bar{CdH_{n-1}L_{n}NO_{3}}$ for the extracted species, the bar denotes species in the membrane phase, and *n* = 1 and 2, depending on the nature of the ionic medium. From the extraction equilibrium constant, *K_ext_*, defined by
(5)$$K_{ext}=\frac{\left[\bar{CdH_{n-1}L_{n}NO_{3}}\right]\left[H^{+}\right]}{\left[Cd^{2+}\right]{\left[\bar{HL}\right]}^{n}\left[NO_{3}^{-}\right]}$$
it is possible to write
(6)$$logD=logK_{ext}+pH+\log\left[NO_{3}^{-}\right]+nlog\bar{\left[HL\right]}$$
once the distribution coefficient, *D*, is considered
(7)$$D=\frac{\left[\bar{Cd\left(II\right)}\right]}{\left[Cd\left(II\right)\right]}$$
where $\left[\bar{Cd\left(II\right)}\right]\;$ and [Cd(II)] are total membrane and aqueous phases equilibrium concentrations, respectively. 

From the plot $logD=f{\left(\bar{\left[HL\right]}\right)}_{pH}$ a value of *n* ≈ 2. Such results perfectly agree with that reported by Aguilar et al. [42] for the extraction of the analyte with Kelex 100 in a solvent extraction system, using kerosene as solvent in nitrate medium. The determined extraction constant is $logK_{ext}=0.02$.

### 2.5. Multivariate Regression Analysis

#### 2.5.1. PLS modeling of VIS and FTIR data

As it was observed that VIS and FTIR information varied with cadmium concentration (Figure 4A,B), the obtained spectral data were submitted to PLS regression analysis to correlate the two data matrices, the X matrix (the VIS and FTIR spectra) and Y matrix (the property, i.e., cadmium content).

The employed concentration range was selected so that a Langmuir-type absorption of the metal ion by the PIM and a favorable preconcentration factor were attained. In the beginning, the complete spectral range was employed. However, from the analysis of the regression coefficients and model parameters, an improvement in the regression parameters was observed when the FTIR range was restricted to 700–410 cm^−1^, and, consequently, further processing was performed using this interval. In the first instance, it was verified that all samples were representative of the same population, using a Hotelling T^2^ test, in conjunction with an F-residual plot. Once no outliers were detected, the analyses results were interpreted. From Figure 5A it was observed that 96% of variability in the VIS spectra was explained using 2 factors; similarly, in the case of FTIR spectra (Figure 5B), the variability explained by the two first factors almost reached 100% for the spectral data.

While the model for VIS data required just two latent variables, the model for FTIR data incorporated a third one, as selected according to a leave-one-out cross-validation process. This result was a direct consequence of the differences in complexity between the VIS and FTIR spectra. The accuracy of the models was quantitatively measured through the RMSEC, the RMSECV, and the slope, intercept, and determination coefficient (R^2^) from the reference vs. predicted values of the property during calibration and cross-validation (Figure 6A,B and Table 2).

As observed good performance was accomplished by the models and no systematic variations were detected based on the slope ($b_{1}$) and intercept values ($b_{0}$) of the regression equations, i.e., the joined F-test for both statistical parameters gave no significant differences at 95% confidence between these values and the expected ones for the slope ($\beta_{1}=1$) and intercept ($\beta_{0}=0$) (*p*-values were 0.9590 and 0.3703 for VIS data for calibration and cross-validation, respectively, and 0.9733 and 0.2731 for FTIR data for calibration and cross-validation, respectively) according to the statistics:(8)$$F=\frac{{\left(\beta_{0}-b_{0}\right)}^{2}+2\bar{x}\left(\beta_{0}-b_{0}\right)\left(\beta_{1}-b_{1}\right)+\left(\sum^{\;}x_{i}^{2}/n\right){\left(\beta_{1}-b_{1}\right)}^{2}}{2S_{e}^{2}/n}$$
where $S_{e}^{2}$ error mean square; *n*, number of data points; $\;\bar{x}$, mean of the reference values; and $\sum^{\;}x_{i}^{2}/n$, mean sum of squares of the reference values. The high values of the determination coefficients give information about the goodness of fit of the models, as this parameter is a statistical measure of how well the regression line approximates the real data points. In addition, the CV-determination coefficients showed a good predictive ability. The agreement between model predictions and ideal behavior is clearly seen in Figure 6A,B from the closeness of the data with the ideal reference line. The regression coefficients (Figure 7A,B) showed similarities with spectral data, which was simpler with VIS than with FTIR data. While the VIS regression coefficient showed a curve profile with a maximal contribution typical of the 8-hydroxyquinoline-Cd(II) complex [54], the FTIR regression coefficient profile included Kelex 100 characteristic IR vibrational bands at about 687 cm^−1^ and 459 cm^−1^, related to C–H out-of-plane bending and to C–O in-plane bending, respectively [55], as expected from the coordination properties of the extractant through its oxygen group.

#### 2.5.2. Figures of Merit of the MID-FTIR-PLS PIM-Based Sensor

At this point it is important to mention that both models gave very good results, being slightly better for FTIR than for the VIS data, as indicated by the higher values of R^2^ and lower values of RMSEC, RMSECV. Based on these observations, the MID-FTIR model was further characterized to extend its application to complex natural waters. Due to the specific spectroscopic signals between the metal ion and the extractant, a minimal or no effect caused by the impurities and the suspended particles is expected, and an accurate analysis could be performed. The unnecessary use of a chromophore agents is an additional advantage of the method. In Table 3, the figures of merit of the FTIR-developed model are presented (linearity, evaluated from $RMSEC=\sqrt{\frac{\sum_{i=1}^{n}{\left(y_{i}-{\hat{y}}_{i}\right)}^{2}}{n-1}}$, cross validation RMSE (RMSECV), determination coefficient (R^2^), cross-validation R^2^ (CV-R^2^), slope, and intercept of the cadmium (II) observed vs. predicted results; sensitivity, as $sen=||s_{k}^{*}||=\frac{1}{||b||}$; analytical sensitivity as $\gamma =\frac{sen}{\left|\delta x\right|}$; minimum discernible concentration difference$,\;\gamma^{-1}=\frac{\left|\delta x\right|}{sen}$; and limits of detection and quantitation ($LD=3.3\delta x\frac{1}{sen}$ and $LQ=10\delta x\frac{1}{sen}$, respectively), where $y_{i}$ and ${\hat{y}}_{i}$ are the estimated and reference values, respectively, of the I, sample, *n* the total number of samples, $||s_{k}||\;$stands for the norm of the sensitivity coefficients of the spectra containing the analyte k at unit concentration and $||s_{k}^{*}||\;$for that corresponding to its *NAS*, $NAS_{i}=\left(x_{i}\cdot b\right)\cdot {\left(b^{T}\cdot b\right)}^{-1}\cdot b^{T}$ where $x_{i}$ is a sample spectrum after preprocessing and b is a column vector of the PLS regression coefficients, ||b|| is the norm of the vector of regression coefficients of the calibration model, and δx is the instrumental noise [56,57,58]. Overall, good performance characteristics were observed. The low value of multivariate selectivity (4.03%) was anticipated according to the employed experimental conditions, as this parameter was a measure of the fraction of the spectrum that was related to the cadmium content in the PIM, and it should be considered that the Kelex 100: cadmium(II) ratio in the PIM was very high (approximately, a ratio of 21). 

#### 2.5.3. Application of the MID-FTIR-PLS PIM-Based Sensor

As the model’s accuracy was dependent on the presence of interferences, the application of the model to three different natural waters (tap, bottle, and pier water) representing complex matrices due to the presence of different ions (i.e., calcium, magnesium, sodium, potassium, chloride, nitrate, sulfate, bicarbonate, fluoride, among others [59,60]) and particulates (e.g., dissolved organic compounds) was evaluated. Samples were spiked with different analyte concentrations and the results of the FTIR method was compared with F-AAS analysis. As observed from Table 4, the non-specific character of 8-hydroxyquinoline, i.e., Kelex 100, which could form complexes with Na (I), Ca (II), and Mg (II), among other ions [61,62], was perfectly compensated by the Cd (II)-Kelex 100 complex specific information contained within the analyzed FTIR spectral region and the pH value selected for the analysis. The joined F-test for the slope and intercept values of the regression equation between the reference vs. the determined values gave no significant differences at the 95% confidence between these values and the expected ones (*p*-value = 0.0774). Consequently, comparable results were obtained between both methods, even in the analysis of a challenging medium like pier water. 

One major advantage of the developed MID-FTIR-PLS PIM-based method was that it did not require the presence in the membrane of a chemical reagent with special properties, either a chromophore species that is able to complex the metal ion, i.e., acting as ionophore [19], or a mixture of an ionophore and a chromophore in the same PIM [17], or a fluorescent reagent [18]. Consequently, there was no need to optimize the PIM composition for chromophore/ionophore/support compatibility [25], so that in practice the methodology is transferable to any PIM system reported up till now. Future work will be addressed toward extending the range of application of the methodology to—(i) a lower analyte concentration range by a careful selection of the dielectric nature of the medium and the dipole moment of the bond associated with IR vibrations of the extracted complex (variation in PIM component’s composition and nature); the larger the dipole moment change and the smaller the position change of the atoms (i.e., of bond lengths or bond angles), the higher the band intensities [63]; and (ii) different analytes, either alone or in a mixture, to fit the purpose of environmental monitoring. In this regard, taking into account a similar behavior of Kelex 100 to its parent structure 8-hydroxyquinoline (logK_extraction_: Cu^2+^ (1.77) > Ni^2+^ (−2.18) > Zn^2+^ (−2.41) > Co^2+^ (−3.7) > Cd^2+^ (−5.29) > Mg^2+^ (−15.13) > Ca^2+^ (−17.89) [64], it is evident that the influence of the presence of other heavy metal ions is a challenge to handle. Chemometric selectivity based on specific absorption bands for the different metal ions might represent a promising alternative to be investigated in future applications. Overall, this article showed the potentiality of the proposed methodology and allowed a proof of the concept for the target purpose. 

## 3. Materials and Methods 

In PIM preparation, Kelex 100 (Sherex Chemical Co. Inc. Dublin, Ohio, USA), cellulose triacetate (CTA, Honeywell Fluka, Charlotte, N.C., USA), and 2-nitrophenyl octyl ether (NPOE, ≥99.0% Honeywell Fluka, Charlotte, N.C., USA) were used as extractants, support, and plasticizer, respectively, using dichloromethane (Merck, Kenilworth, N.J., USA) as a casting solvent. Working solutions were prepared from tetrahydrated cadmium (II) nitrate (≥99.0% Fluka) while a 1000 mg/L Sigma-Aldrich AAS standard solution was diluted using deionized water, for the preparation of the standards for F-AAS determinations. Tris(hydroxymethyl)aminomethane (TRIS, 99.8% Aldrich, pH 7–9), 4-morpholino ethanesulfonic acid (MES, 99.5% Sigma, pH 5.5–6.7), sodium acetate (99% Aldrich Chem. Co., St. Louis, MO, USA) / acetic acid (99.7%, Sigma-Aldrich) buffer solution (pH 3.8–5.8), and hydrochloric acid (37% Sigma-Aldrich Chem. Co., St. Louis, MO, USA) were employed to adjust the pH of the aqueous solutions.

Extraction experiments were carried out with a model 75 Wrist ActionTM shaker (Burrell Scientific Inc, Pittsburgh, Pa., USA). The spectrometers Perkin Elmer 3100, Perkin Elmer Lambda 2 and Perkin Elmer Spectrum GX (Waltham, Mass., USA) were used for F-AAS, VIS, and FTIR determinations, respectively. A Metrohm 620 pH-meter (Herisau, Switzerland) was employed for pH measurement and adjustment. A Fowler IP54 micrometer (Fowler High Precision, Newton, Mass., USA) was used for measuring PIM thickness. The Unscrambler 10.5.1 software (Camo Analytics, Oslo, Norway) was employed for PLS analyses.

PIMs were prepared by dissolving the weighted amounts of CTA, NPOE, and Kelex 100 in dichloromethane. The mixture was stirred for 1 h on a magnetic plate with a stirring bar. The solution was then casted in a 5 cm diameter Petri dish and rested for 24 h for solvent evaporation. Finally, the membrane was carefully peeled off and the whole piece was used in all experiments. PIMs were transparent films with an average thickness of (46 ± 11) µm, average weight of (112.3 ± 0.0056) mg and an average diameter of (4.89 ± 0.0322) cm.

As for the biphasic solid-liquid extraction experiments, the membranes were introduced in 50 mL polypropylene Falcon tubes, in the presence of 30 mL of aqueous solution, with cadmium (II) at fixed concentrations. The tubes were shaken for regular time intervals and the aliquots of 400 µL were taken and diluted to 2 mL before F-AAS analysis, using the conditions recommended by the manufacturer (λ 228.8 nm, 7 nm slit, air/acetylene flame). Experiments were performed on a duplicate basis with an average RSD of 5%. No analyte elution step was required as direct analysis of the PIMs was performed. This was opposite to the traditional three-phases configuration (feed, membrane, strip) usually employed for metal ion removal. The employed set-up was commonly found when the PIMs were used for sensing in chemical analysis [25].

Chemometric analyses were conducted using a training set of 15 different concentrations (in duplicate), ranging from 6.94 × 10^−7^ to 3.82 × 10^−4^ mol dm^−3^. The concentration of the calibration standards was determined by F-AAS (Perkin Elmer 3100, Waltham, Mass., USA). The same PIMs were analyzed by VIS and MID-FTIR spectroscopies. VIS spectra were recorded in transmission mode by triplicates, in the range 500–390 nm, after sandwiching the membrane between two Petri dishes to avoid wrinkles and movement. FTIR spectra were recorded in transmission mode by triplicates in the range of 4000–400 cm^−1^. The PIM was mounted in the transmission accessory of the equipment and scanned 45 times to record the spectrum. The six spectra for each concentration in VIS and IR modes were then averaged for multivariate analysis, using the spectra calculator of the software. The best results were obtained after mean-centering the spectra and, in the case of the FTIR data, applying the unit vector normalization. Chemometric analyses were validated through cross-validation procedures. An in-house made MATHLAB program was used for the figures of merit determination, using the outputs of the Unscrambler 10.5.1 (Camo Analytics, Oslo, Norway) software. The applicability of the method was tested by spiking with reference concentrations bottle, tap, and pier (Cuemanco, Xochimilco, Mexico) water, after filtration of the samples.

## 4. Conclusions

The results showed that sorption coupled with direct spectroscopic analysis, using a PLS chemometric approach in a PIM, constituted a potential sensor for metal ion determination in waters with results comparable to F-AAS. On-site analysis with portable equipment was anticipated through the proof of the concept of the methodology in the case of the measurement of cadmium (II) in aqueous solutions, using the commercial extractant Kelex 100 immobilized in the membrane. MID-FTIR showed to be an adequate technique for cadmium (II) analysis in the membrane, once the metal was preconcentrated, which was barely used for quantitative purposes in this type of application. The specificity of bonding between the metal and the extractant, together with the optimization of the uptake conditions, allowed the selectivity of the system toward competing ions, while the chemometric treatment of the spectral data allowed the selectivity of the metal signal toward the organic components conforming the membrane. A wide range of future applications is anticipated to target different metal ions with specific extractants immobilized in PIMs, as no chromophores or fluorescent reagents are needed in this novel type of application. The simplicity of the system is expected to optimize the PIM composition for chromophore/ionophore/support compatibility, and in its transferability, as all organic extractants present active functional groups in the MID-IR region.

## Figures and Tables

**Figure 1 molecules-25-03436-f001:**
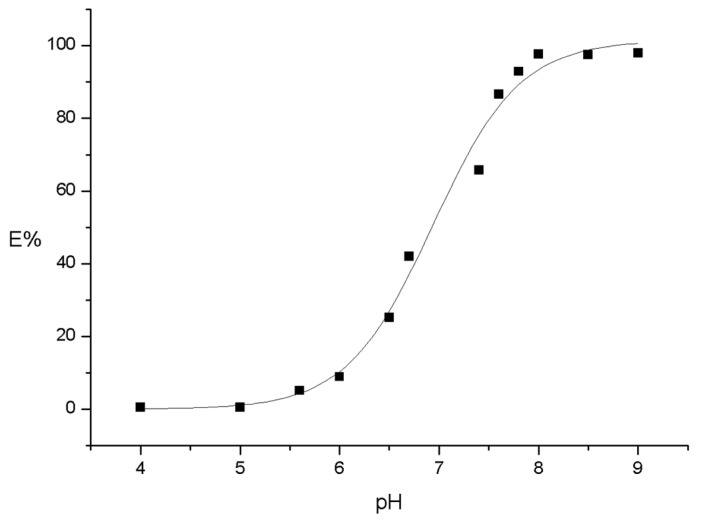
Percent of extraction of cadmium (II) as a function of pH in the PIM. Aqueous phase: 1 × 10^−4^ mol dm^−3^, PIM: 23.4% CTA, 54.5% NPOE, and 21.1% Kelex 100 (*w*/*w*).

**Figure 2 molecules-25-03436-f002:**
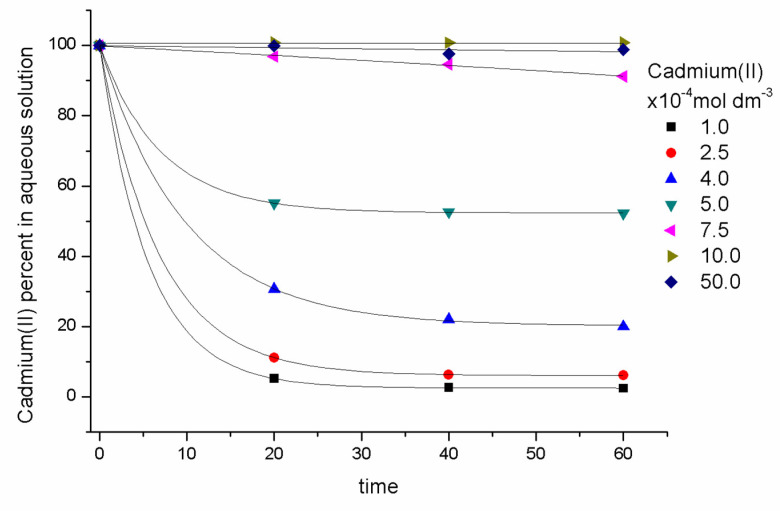
Cadmium (II) concentration profiles in the aqueous phase (pH = 8, 1 × 10^−3^ mol dm^−3^ TRIS) for different initial concentrations. PIM: 23.4% CTA, 54.5% NPOE, and 21.1% Kelex 100 (*w*/*w*).

**Figure 3 molecules-25-03436-f003:**
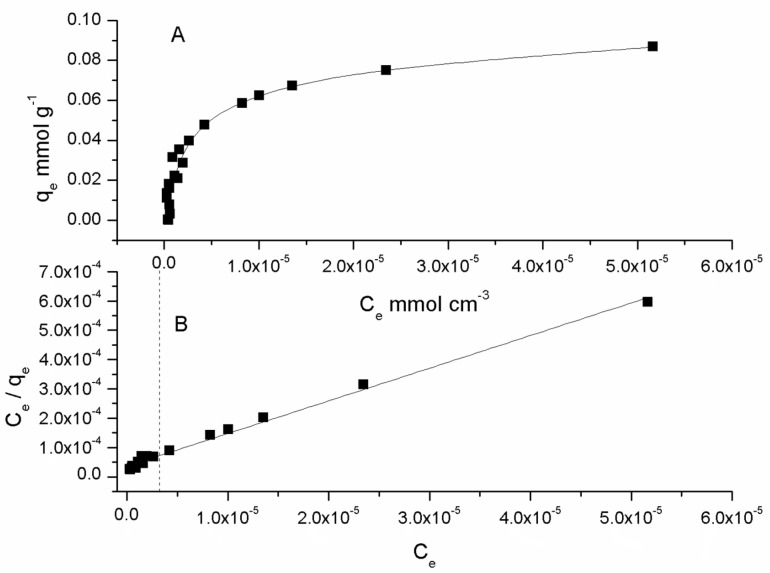
Cadmium (II) adsorption isotherm (**A**) and its linearized form (**B**). Aqueous phase: 6.94 × 10^−7^–3.82 × 10^−4^ mol dm^−3^, pH = 8.0 (1 × 10^−3^ mol dm^−3^ TRIS), PIM: 23.4% CTA, 54.5% NPOE, and 21.1% Kelex 100 (*w*/*w*).

**Figure 4 molecules-25-03436-f004:**
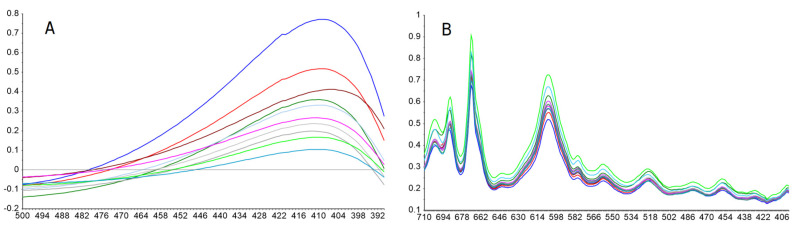
VIS (500–390 nm) (**A**) and MID–FTIR (710–400 cm^−1^) (**B**) spectra of PIMs after equilibration with different cadmium (II) initial concentrations in the aqueous phase (1.63 × 10^−6^–3.82 × 10^−4^ mol dm^−3^) at pH = 8.0 (1 × 10^−3^ mol dm^−3^ TRIS), PIM: 23.4% CTA, 54.5% NPOE, and 21.1% Kelex 100 (*w*/*w*).

**Figure 5 molecules-25-03436-f005:**
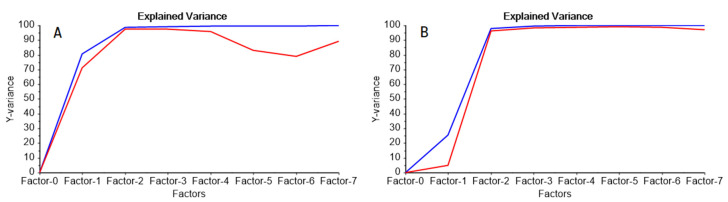
Percent of spectral variance explained by the VIS (**A**) and MID-FTIR (**B**) models as a function of the number of latent variables for calibration (blue) and cross-validation (red).

**Figure 6 molecules-25-03436-f006:**
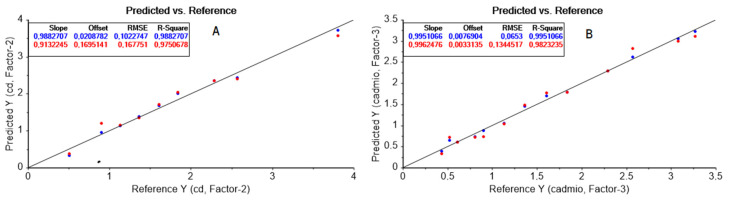
Cadmium (II) observed vs. predicted results for VIS (**A**), and MID–FTIR (**B**) data showing the fitting parameters during calibration (blue) and cross-validation (red). The reference line with zero intercept and with a slope of one is included.

**Figure 7 molecules-25-03436-f007:**
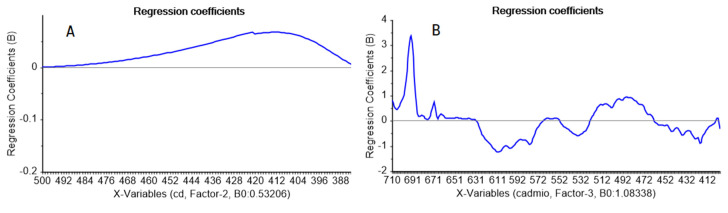
Regression coefficients of the VIS (**A**) and MID-FTIR (**B**) analyses as a function of wavelength (500–390 nm) or wave number (710–400 cm^−1^), respectively.

**Table 1 molecules-25-03436-t001:** Parameters obtained from the linearized form of the absorption isotherms and ANOVA test results.

Parameter	*C_e_* = 1.11 × 10^−6^ to 9.33 × 10^−7^	*C_e_* = 1.63 × 10^−6^ to 5.16 × 10^−5^
*K_L_* [cm^3^ mmol^−1^]	17.028	2.133
*q_max_* [mmol g^−1^]	0.034	0.092
R^2^	0.8421	0.9959
R_L_	0.005–0.008	0.01–0.03
CV-R^2^	0.7539	0.9834
Model sum of squares	2.37 × 10^−9^	1.71 × 10^−7^
Error sum of squares	7.36 × 10^−10^	2.56 × 10^−10^
Model mean square	2.37 × 10^−9^	1.71 × 10^−7^
Error mean square	9.20 × 10^−11^	6.41 × 10^−11^
F-value	25.81	2665.23
*p*-value	9.53 × 10^−4^	8.42 × 10^−7^

**Table 2 molecules-25-03436-t002:** Statistical parameters associated with the accuracy of the developed methods.

Parameters	VIS Calibration	VIS Validation	MID-FTIR Calibration	MID-FTIR Validation
RMSEC	0.1022		0.0653	
RMSECV		0.1677		0.1344
Slope	0.9882	0.9132	0.9951	0.9962
Intercept	0.0208	0.1695	0.0076	0.0033
R^2^	0.9882	0.9750	0.9951	0.9823

**Table 3 molecules-25-03436-t003:** Analytical figures of merit for the developed MID-FTIR method.

Figure of Merit	Value
RMSEE	0.0653
RMSECV	0.1344
R^2^	0.9951
CV-R^2^	0.9823
slope	0.9951
intercept	0.0076
range	(0.45–3.27) × 10^−4^ mol dm^−3^
sen	0.0164 mol^−1^ dm^3^
γ	7.26 mol dm^−3^
$\gamma^{-1}$	0.14 mol^−1^ dm^3^
*Mean selectivity*	0.0403
LD	0.45 × 10^−4^ mol dm^−3^
LQ	1.37 × 10^−4^ mol dm^−3^

**Table 4 molecules-25-03436-t004:** Results of the analysis of cadmium (II) in real water samples spiked with the analyte.

Sample	Reference Value (F-AAS) × 10^4^ mol dm^−3^	Determined Value (MID-FTIR) × 10^4^ mol dm^−3^
Tap	3.10 ± 0.10	3.25 ± 0.25
	1.00 ± 0.10	1.35 ± 0.22
	0.56 ± 0.10	0.52 ± 0.35
Bottle	0.87 ± 0.10	1.08 ± 0.20
	0.51 ± 0.10	0.52 ± 0.30
Pier	2.75 ± 0.10	3.08 ± 0.25
	0.96 ± 0.10	1.07 ± 0.20
	0.56 ± 0.10	0.47 ± 0.30

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
