# Peer review of "Determination of Cadmium (II) in Aqueous Solutions by In Situ MID-FTIR-PLS Analysis Using a Polymer Inclusion Membrane-Based Sensor: First Considerations"

_molecules, 2020, doi:10.3390/molecules25153436_

Round 1

Reviewer 1 Report

A method to detect the content of cadmium (II) in aqueous solutions was proposed by using polymer inclusion membrane as a VIS and MID-FTIR sensor and PLS modeling. The experiment and calculations were described in detail. The paper can be accepted for publication, but revisions are still needed.

(1) In the past decades, many works have been reported using absorbent (including membrane) as a sensor and chemometric modeling as an analyzing tool for spectroscopic analysis. In these works, it may be more concentrated on NIR spectroscopy. An introduction of these works is needed, and it may be better to comment on the difference of NIR and MID-FTIR.

(2) The description of PLS is not necessary, but the details of the usage in this study should be described.

(3) To show the practicability of the proposed method, samples with complex matrix (interferences) should be tested. We cannot believe the good result obtained by chemometric modeling of the spectra changing with one factor.

Author Response

CONCEPTUAL/EXPERIMENTAL COMMENTS

(Obs) Line 100: please specify how the pH is regulated

(R) The buffers solutions employed are clearly specified now in lines 362-363 of the revised version in the Materials and Methods section.

(Obs) Line 126: ERRATA: “KL (dm3 mg-1),” CORRIGE: “KL (cm3 mmol-1)”. The correction is needed, since Ce in measured in (mmol cm-3,) and the product of KL and Ce must be adimensional, being summed to an adimensional quantity (“1”).

(R) Corrected

(Obs)Table 1: please add the ANOVA test for the model, and particularly the p-value for the F parameter (ratio of Model Sum of Squares to Residual Sum of Squares, divided by the corresponding degrees of freedom).

(R) Done (see Table 1)

(Obs)Line 128: referring to the sentence “The initial amount of cadmium (II) in the aqueous phase can be predicted”, I recommend to calculate, for each univariate model, not only the regression coefficients (descriptive ability) but also the cross-validation regression coefficients (predictive ability). This is correctly done by the Authors for the multivariate model (e.g. figure 6), but it is important also for a univariate regression (model y = a + b x). The univariate CV-correlation-coefficients could be added to Table 1.

(R) Done (see Table 1)

(Obs) Lines 178-179: when numeric values are reported without specifying the relevant uncertainty, I recommend to use the same numeric format, for instance 3 significant digits; so, 10.86 becomes 10.9 and 2.8 becomes 2.80 or 2.81 or 2.82 or 2.83 or 2.84 depending on experimental results. Otherwise, the reader understands that the precision for 2.8 is 1/28=3.57%, while the precision for 10.86 is 1/1086=0.0921%, which is incoherent.

(R) Corrected (lines 194-195 revised version)

(ObsR)Table 2: please, use a uniform format for numbers; three significant digits could be good. If 3 digits are visualized, 0.1022 becomes 0.102; 0.0076 becomes 0.00760 or 0.00761 or 0.00762 or 0.00763 or 0.00764, etc.

(R) Corrected.

Lines 253-255. “The high values of the determination coefficients give information about the goodness of fit of the models, as this parameter is a statistical measure of how well the regression line approximates the real data points.” This is true, and it is referred to the blue numbers in figure 6; however, also the red numbers deserve attention: I recommend to underline that these CV-determination-coefficients show a good predictive ability.

(R) Done (lines 278-279 revised version)

 EDITING ERRORS

(Obs)Line 12: ERRATA: “method” CORRIGE: “methods”

(R) Corrected.

(Obs)Line 47: ERRATA: “However difficulties” CORRIGE: “However, difficulties”

(R) Corrected.

(Obs)Line 56: ERRATA: “consumption liquid membrane-based sample preparation and preconcentration techniques has gained growing attention” CORRIGE: “consumption, liquid membrane-based sample preparation and preconcentration techniques have gained growing attention”

(R) Corrected.

(Obs)Line 271: ERRATA: “which” CORRIGE: “whose”

(R) Corrected.

(ObsR)Lines 270-276: too long sentence, it is hard to well understand. Please, modify it.

(R) Modified (lines 297-300 revised version)

(ObsR)Table 3: I suggest to remove the formulas and move them in the text. Add also the cross-validation R-square. Please use always the same format for the regression coefficients: “R” or “r”.

(R) Done (see Table 3). The formulas are within the text in lines 301-311 of the revised version.

Reviewer 2 Report

The manuscript molecules-853645,  entitled “Determination of cadmium (II) in aqueous solutions by in situ MID-FTIR-PLS analysis using a polymer inclusion membrane-based sensor: first considerations”,  is a good work.

Only minor revisions are needed, and then, in my opinion, the paper will be ready for publication.

Author Response

(Obs) In the paste decades, many works have been reported using sorbent (including membrane) as sensors and chemometric modelling as an analyzing tool for spectroscopic analysis. In these works, it may be more concentrated on NIR spectroscopy. An introduction of these works is needed, and it may be better to comment on the difference of NIR and MID-FTIR.

(R) Done. The new paragraph (lines 74-89 of the revised version) addresses this question.

(Obs) The description of PLS is not necessary, but the details of the usage in this study should be described.

(R) Done. The model was removed and more a more detailed explanation was included in the Materials and Methods section.

(Obs)To show the practicability of the proposed method, samples with complex matrix (interferences) should be tested. We cannot believe the good result obtained by chemometric modelling of the spectra changing with one factor.

(R) In lines 317-321 of the revised version the phrase “As the model’s accuracy is dependent on the presence of interferences, the application of the model to three different natural waters (tap, bottle and pier water) representing complex matrices due to the presence of different ions (i.e., calcium, magnesium, sodium, potassium, chloride, nitrate, sulfate, bicarbonate, fluoride, among others [50, 51]) and particulates (e.g. dissolved organic compounds) was evaluated” to emphasize that the method was applied in these complex natural samples with excellent results as shown in Table 4. As we are aware that not all possible interferences were evaluated in lines 337-350 future directions in this sense are presented based on the extraction behavior of the extractant.

Reviewer 3 Report

Manuscript title: Determination of cadmium (II) in aqueous solutions by in situ MID-FTIR-PLS analysis using a polymer inclusion membrane-based sensor: first considerations

The study presents an interesting approach about the use of PIMs as sensors for metal ions. The fact that the metal can be measured directly from the PIM without needing for an additional elution step is very promising. However, this work presents a clear drawback that is, the high metal concentration (in the range 10-100 mg L-1) investigated which limits its application for real situations. Then, my first point is why lower concentrations were not studied? Maybe, by tuning the PIM composition, better limits of detection could be achieved.

Then, the second point is that the information given in the experimental part is rather scarce, which makes difficult to figure out the experimental setup. Do the authors use the whole PIM for each experiment? How was the setup for UV and FTIR measurements? Do they use fibre optics? And what about the measurement mode? Also, information about the experimental conditions should be added in the figure captions.

Other comments:

Line 96: replace m/m by w/w

Lines 102-104: the results in Figure 1 are explained because of the speciation of Cd in aqueous solution. Accordingly, the extraction should be higher at lower pH where the free Cd(II) species predominates. Obviously, this is not the case. Why?

In Figure 3, Y-axis, what is the meaning of g in mmol g-1? Do they refer to the mass of PIM or the mass of Kelex in the PIM? This also applies to Table 1.

Lines 152-157: The values of capacities given deal basically with biosorbents. It would be of interest seeing how the obtained values compare with other synthetic materials.

In equation (5), the units demoting the concentration of Cd(II) in the membrane should be indicated.

Lines 172-176: Are these membrane systems a type of three-phase membrane design where the metal is recovered in an aqueous stripping phase? This point should be discussed.

Figure 5, Figure 6, Figure 7. The axis title can hardly be read: Please use a larger font size.

Line 280:”the Kelex 100:cadmium(II) ratio in the PIM is very high”: can you give a number for that?

Line 311: should be Cu2+(1.77)> Ni2+ (-2.18)

Finally, the author state that this type of sensor can be considered a better approach for Cd(II) sensing, in terms of simplicity, compared to the usual optical sensor based on the combination of a specific ionophore plus a chromoionophore. A reference is needed here and the idea should be further discussed in the final part of the work.

Author Response

(Obs) […] This work presents a clear drawback, that is the high metal concentration (in the range 10-1000 mg L-1) investigated in which limits its application for real situations. Then, my first point is why lower concentrations were not studied? Maybe, by tuning the PIM composition, better limits of detection could be achieved.

In lines 317-321 of the revised version the phrase “As the model’s accuracy is dependent on the presence of interferences, the application of the model to three different natural waters (tap, bottle and pier water) representing complex matrices due to the presence of different ions (i.e., calcium, magnesium, sodium, potassium, chloride, nitrate, sulfate, bicarbonate, fluoride, among others [50, 51]) and particulates (e.g. dissolved organic compounds) was evaluated” to emphasize that the method was applied in these complex natural samples with excellent results as shown in Table 4. As we are aware that not all possible interferences were evaluated in lines 337-350 future directions in this sense are presented based on the extraction behavior of the extractant. In the same lines a possible solution to attain a lower limit of detection is given. As mentioned in the text, we are aware about the afore mentioned limitation of the work; however, up to our knowledge, this is the first time in which the proposed methodology is employed, i.e., no reference concerning the detection capabilities of the method were available at the time of planning the experimentation, and the performance characteristics measured (Table 3) so that future work will be performed to challenge the application. For this reason, the title of the manuscript highlights “first consideration” as an initial stage of research in this type of systems.

The second point is that the information given in the experimental part is rather scarce, which makes difficult to figure out the experimental setup.

The whole manuscript was revised to get a better understanding of the experimental work done.

(Q) Do the authors use the whole PIM for each experiment?

R: Yes. This is now indicated in line 372 of the revised version.

(Q) How was the setup for UV and FTIR measurements?

R: Lines 385-395:

Chemometric analyses were conducted using a training set of 15 different concentrations (in duplicate) ranging from 6.94x10-7 to 3.84x10-4 mol dm-3. The concentration of the calibration standards was determined by FAAS. The same PIMs were analyzed by VIS and MID-FTIR spectroscopies. VIS spectra were recorded in transmission mode by triplicate in the range 500- 390 nm, after sandwiching the membrane between two Petri dishes to avoid wrinkles and movement. FTIR spectra were recorded in transmission mode by triplicate in the range 4000-400 cm-1. The PIM was mounted in the transmission accessory of the equipment and scanned 45 times to record the spectrum. The six spectra for each concentration in VIS and IR modes were then averaged for multivariate analysis using the spectra calculator of the software. Best results were obtained after mean-centering the spectra and, in the case of FTIR data, applying unit vector normalization. Chemometrics analyses were validated through cross-validation procedures.

(Q) Do they use optic fiber? And what about the measurement mode?

R: see above

(Obs) Also, Information about the experimental conditions should be added in the figure captions.

R: They were included.

(Obs) The results in Figure 1 are explained because of the speciation of Cd in aqueous solution. Accordingly, the extraction should be higher at lower pH where the free Cd(II) species predominates. Obviously, this is not the case, why?

R: This behavior can be explained considering the acid nature of the extractant (Kelex-100). Lines 102-107 address this question.

(Q) In Figure 3, Y-axis, what is the meaning of g in mmol g-1? Do they refer to the mass of PIM or the mass of Kelex in the PIM? This also applies to Table 1.

R: The units mmol g-1 of the variable qe represent mol of Cd(II) per gram of membrane, while the units of the variable Ce (mmol cm-3) represent the metal concentration in solution at equilibrium.

(Obs) The values of capacities given deal basically with biosorbents. It would be of interest seeing how the obtained values compare with other synthetic materials.

R: They were included in lines 167-168 of the revised version.

 (Obs) In equation (5), the units denoting the concentration of Cd(II) in the membrane should be indicated.

R: They were included in the text following the equation.

(Q) Are these membrane systems a type of three-phase membrane design where the metal is recovered in aqueous stripping phase? This point should be discussed.

R: Lines 379-383 were added to address this question: “No analyte elution step was required as direct analysis of the PIMs was performed. This is opposite to the traditional three-phases configuration (feed, membrane, strip) usually employed for metal ion removal. The employed set-up is commonly found when PIMs are used for sensing in chemical analysis [25]”.

(Obs) Figure 5, Figure 6, Figure 7 axis title can hardly be read: Please use a larger font size.

R: The ranges of the variables were included in the captions. As the format of the journal is only on-line, it is possible to zoom-in all the figures satisfactorily.

(Obs) The “Kelex-100:  Cd (II) ratio in the PIM is very high”: can you give a number for that?

R: Now it is defined in line 315.

(Obs) Should be Cu2+ (1.77) > Ni2+ (-2.18).

R: Corrected.

(Obs) The author states that this type of sensor can be considered a better approach for Cd(II) sensing, in terms of simplicity, compared to the usual optical sensor based on the combination of a specific ionophore plus a chromoionophore. A reference is needed here, and the idea should be further discussed in the final part of the work.

R: The required reference was added and the phrase “Simplicity of the system is expected in terms of no need to optimize the PIM composition for chromophore/ionophore/support compatibility, and in its transferability, as all organic extractants present active functional groups in the MID-IR region” added to the conclusions.

Round 2

Reviewer 1 Report

The manuscript was revised according to the comments of the reviewers. The paper can be accepted for publication, but I still suggest citing more literatures of the similar works in NIR spectroscopic analysis, such as Talanta, 2009, 79:339-343; 2010, 82, 1802–1808; 2011, 84(3): 679–683; Analytica Chimica Acta, 2010, 670, 39-43; etc.

Author Response

Thank you. Now more relevant literature to the theme was included.

Reviewer 3 Report

Just some minor corrections:

In figure caption of Figure 3, the metal concentration in the aqueous phase was varied between 6.94x10-7 to 3.82x10-4 mol dm-3 and not fixed at 1x10-4 mol dm-3. The same occurs in figure caption for Figure 4.

Author Response

Thank you. The labels were corrected.